# Digitization in Everyday Nursing Care: A Vignette Study in German Hospitals

**DOI:** 10.3390/ijerph191710775

**Published:** 2022-08-30

**Authors:** Lisa Korte, Sabine Bohnet-Joschko

**Affiliations:** Chair of Healthcare Management and Innovation, Faculty of Management, Economics and Society, Witten/Herdecke University, 58455 Witten, Germany

**Keywords:** digitization, digitalization, digital health, hospital, nursing, nurses, vignette experiment

## Abstract

(1) Background: Digitization in hospital nursing promises to transform the organization of care processes and, therefore, provide relief to nurse staffing shortages. While technological solutions are advanced and application fields numerous, comprehensive implementation remains challenging. Nursing leadership is crucial to digital change processes. This vignette study examined the effects of the motives and values on nurses’ motivation to use innovative technologies. (2) Methods: We asked hospital nurses in an online vignette study to assess a fictitious situation about the introduction of digital technology. We varied the devices on the degree of novelty (*tablet/smart glasses*), addressed motives (*intrinsic/extrinsic*), and values (*efficiency/patient orientation*). (3) Results: The analysis included 299 responses. The *tablet* vignettes caused more motivation than those of the *smart glasses* (Z = −6.653, *p* < 0.001). The dataset did not show significant differences between *intrinsic* and *extrinsic motives*. The nursing leader was more motivating when emphasizing *efficiency* rather than *patient orientation* (Z = −2.995, *p* = 0.003). (4) Conclusions: The results suggest *efficiency* as a motive for using known digital technologies. The nursing staff’s willingness to use digital technology is generally high. Management actions can provide a structural framework and training so that nursing leaders can ensure their staff’s engagement in using also unknown devices.

## 1. Introduction

The impact of digitization on healthcare is profound and involves comprehensive change processes entailing health professionals’ willingness to enhance new competencies [1,2,3]. Hospitals, as complex expert organizations, have well-established information and communication technology structures. Thus, they are often the starting point for further technological development, research, and implementation [4,5]. The field of digital applications ranges from electronic documentation to robot-assisted procedures and can increase the quality of care, patient safety, and better patient outcomes, as well as efficiency and cost-effectiveness [6]. Health system challenges arising from demographic change and increasing demand for health care services while facing a shortage of skilled professionals are hoping to be eased by digitization [5,6]. New technologies can address specific existing problems: nurses spend at least thirty percent of their working day filling out forms and documenting patient data. Digitized documentation processes reduce paperwork, create optimized schedules, and increase productivity, allowing nurses more time for patient care [7,8,9,10]. Electronic medical records encourage health professionals’ exchange and decision-making [11,12]. Another digital opportunity to support patient safety and quality of care is video consultation [13], which nurses in hospitals can perform, e.g., with tablets. However, not all digital opportunities are better or even more suitable options. The ability to use digital innovations is not at the same time the necessity of their application. Eventually, the usage of technologies has to face problems in nursing, such as time pressure and work compaction. Nurses are the largest group of healthcare professionals in hospital settings [14]. The successful digitization in hospitals crucially needs nurses’ engagement and motivation to embrace technological change and to reach the previously listed advantages [6,15,16]. Nursing leaders in hospital departments can expedite change by addressing *intrinsic* and *extrinsic motives* (*IM* and *EM*) as well as important values, such as *patient orientation* and *efficiency orientation* (*PO* and *EO*) [17,18,19,20,21].

### Theoretical Framework, Aims, and Hypotheses

Focusing on leadership communication [22,23,24], the aim of this study was to examine how nursing leaders in hospitals can promote the implementation of digital technologies in their teams and departments. What motives and values can they address to support nurses’ motivation to use different digital innovations? Based on systematic literature research and consultation with nurses and nursing leaders, we selected a *tablet* (*T*) as the incremental innovation. For a higher level of innovation, we discussed nursing robots and smart glasses. As the advisory nurses linked robots immediately with doubts, we decided to choose *smart*
*glasses* (*SG*) [25,26]. Both devices are designed to support care activities, documentation processes, and the subsequent quality of care and patient safety [25,27]. They are valid solutions to existing problems by offering an effective and efficient way of filling out forms and documenting patient data, as well as structuring the care processes and sharing data with nursing colleagues and other involved professionals. Consequently, nurses can use the saved time for more intensive patient care and see more patients [7,8]. Likewise, both devices can support nurses and increase patient safety when nurses use them in the context of video consultation [9,13]. Smart glasses are also promising in connection with nursing education. For instance, they can increase the learning attitude and motivation of nursing students, as well as support students’ learning and comprehension [28,29]. We aimed to understand the differences regarding the level of innovation by offering *tablets* as a well-established technology versus *smart glasses* that are mostly unknown. Using these is difficult to imagine because of the unfamiliar way of wearing them. We developed the following hypotheses: 

**H1.** 
*The use of a T is associated with higher motivation than the use of SG;*


**H2.** 
*IM motivates the nursing staff more to use digital innovations than EM;*


**H3.** 
*PO motivates the nursing staff more to use digital innovations than EO;*


**H4.** 
*The degree of innovation (T/SG) has a higher influence than the values (PO/EO) and motives (IM/EM);*


**H5.** 
*The values (PO/EO) are associated with higher motivation than motives (IM/EM).*


## 2. Materials and Methods

### 2.1. Study and Questionnaire Design

We used an experimental vignette design to examine our hypotheses. The vignette methodology involves a brief description of a fictitious situation, which is varied for specific factors [30,31]. The design is particularly suited for health and social research because it allows the investigation of latent variables, such as attitudes or behavioral intentions [31,32,33,34,35]. The advantage of vignette analysis is the reduction in socially desirable responses by the experimental but realistic design of everyday situations [30]. This strengthens both internal and external validity [33,36,37,38].

Based on an extensive literature review and in-depth consultation with nurses, nursing leaders, and researchers, we developed vignettes to test the influence of motives and values in leadership communication on the motivation to use innovative technologies. We conducted systematic literature research on precise, relevant, motivating leadership characteristics with an effect on staff engagement [39], performance [40], and quality in nursing care [41]. Building on that, consultations with nursing practitioners were crucial to our design of vignettes and choice of devices. We optimized the vignettes and survey in three pretest rounds and finalized the wording. We introduced the vignettes as follows: We asked participants to envision being at a team meeting at their workplace and the nursing leader announcing the forthcoming implementation of a device to support care activities and documentation processes. To distinguish the levels of innovation, we introduced two different devices. We chose *tablets* as the most known device to nurses for incremental innovation and *smart glasses* as the widely unknown device for radical innovation. We explained the devices, their purpose, and usage [4,25,26,42].

We designed the nursing leaders’ announcement of the upcoming change to combine different motives (*IM/EM*) and values (*PO/EO*). In total, there were four different vignettes for each digital device (*T/SG*). We compared the two motives of *IM* and *EM* by developing and testing phrases that indicate *IM* as the motivation for a task that is inherently interesting or enjoyable and *EM* as the type of motivation that is stimulated by an external reward [43,44]. The terms “interesting activity” and “expansion of professional competencies and opportunities” represented *IM* and *EM* [43,44,45,46,47,48]; “more time for individual patients” and “completing tasks more quickly” represented *PO* and *EO* as the widely discussed values in hospital nursing [21,49,50,51,52,53]. The participants were presented with the different vignettes and asked to rate their motivation to use the respective device.

We conducted the vignette study via LimeSurvey, an academic online survey tool. It took the participants approximately 10 min to complete the survey. There were no inclusion criteria except for working as a nurse in a German hospital. Instead, we queried additional characteristics to specify and control influences within the respondent group. The survey started with the vignettes, and we added questions about the particular situation, the sociodemographic data of the person, general willingness to use technology, professional background, job satisfaction, facility, and professional characteristics. Except for an open comment option at the end of the questionnaire, we specified all of the answer options, mainly by rating on a six-point Likert scale.

### 2.2. Recruitment

We distributed the online survey between 24 November 2021 and 20 January 2022. We spread the link across personal social networks, social platforms, and email distribution lists for related staff and experts with further connections. Several hospitals and networks of professionals also distributed the survey.

### 2.3. Data Processing and Statistical Analysis

We removed incomplete and erroneous questionnaires, such as those not filled out by hospital nurses. Since all crucial questions referred to non-sensitive and, therefore, mandatory data, there were no missing values. Other questions had answer options such as “do not know” or “do not know exactly.” We coded the data in text format into numeric indicator variables.

We used two short scales on general willingness to use technology and job satisfaction from the compilation of social science items (“ZIS”) of “GESIS-Leibniz Institute for Social Sciences e.V.” [54,55]. The individual items were coded in the same directions in order to define little willingness to use technology or low job satisfaction with small numbers. For both scales, we calculated new variables to indicate the average values. We categorized the variables with many expressions, such as the age in the groups.

For the evaluation of each vignette, the data distribution was tested. Due to the non-normally distributed variables, we applied the Friedman test as a non-parametric test for the comparison of more than two dependent samples—in this case, the different vignettes. We used the same procedure for recalculating dependent variables in which we considered the individual dimensions and characteristics in isolation. In order to find differences in their influence on the dependent variable of motivation to use, we performed Dunn–Bonferroni tests as post hoc tests. For direct comparisons between two vignettes, we applied the Wilcoxon test for two connected samples. We conducted group comparisons, regression analyses, and correlation calculations to control the association between independent variables and the participants’ motivation to use digital technologies.

## 3. Results

### 3.1. Sample: Respondent Characteristics

After removing 232 incomplete data sets, as well as 13 questionnaires that did not meet the inclusion criteria, 299 complete data sets remained. With a number of 229, three-quarters of the respondents were female (77%). The average age was 37 years. The youngest person was 18, and the oldest was 63 years old. The largest age group was 20 to 29 years old (32%). Most participants had a high school diploma (“Abitur”) (38%) or a university degree (33%) as the highest level of education. Table 1 presents these sociodemographic characteristics.

#### 3.1.1. Sample: Profession and Workplace

Overall, 80% of the participants were part of the general professional health care and nursing group. Almost one-quarter of the respondents had an academic nursing degree (23%). The sample was distributed across various specialties. A high proportion (21%) of participants worked in intensive care medicine.

The average scope of employment was around 85%, with around 60% working full-time. Almost 22% had personnel responsibility. Nearly 60% of the people worked in large hospitals with at least 800 beds. More than two-thirds of the respondents worked at hospitals under public ownership (71%). Table 2 shows the most important data on the workplace.

#### 3.1.2. Sample: Technology Readiness and Job Satisfaction

The relevant values for general willingness to use technology and general job satisfaction describe the following: Most participants had a more positive attitude towards technology. The mode and median were “rather high” (4). In total, the respondents selected the upper values (4 to 6) more frequently, which indicated a greater willingness to use technology (91%). Regarding general job satisfaction, the situation was different. Although the median and mode were also 4, almost half of the respondents selected the three lower values (1 to 3) (Table 1).

#### 3.1.3. Sample: Professional Motives and Values

We queried the general importance of the motives and values addressed in the vignettes at the end of the questionnaire. All of the characteristics (*IM/EM/PO/EO*) were important in most respondents’ everyday working life. The median and mode were higher for *IM* and *PO* (6) than for *EM* and *EO* (5). Table 3 shows an overview of these data.

### 3.2. Motivation to Use Digital Innovations in a Situational Context

Within the *T* vignettes, the Friedman test did not prove any significant differences between the four combinations of motives and values in the nursing leader’s announcement. The mode for the two *extrinsic T*-vignettes was 3 and was thus a little lower than that for the *intrinsic T* vignettes (4). The median of the combination *T-EM-PO* (3) was also lower than the median for the other three *T* vignettes (4) (Table 4). We also found no significant differences within the *SG* vignettes. The median for all four vignettes was 3, but the mode for the combination *EM-PO* was higher (4) compared to those of all the others (3) (Table 4). We finally identified significant differences by analyzing the individual ratings in isolation via the Friedman test (Figure 1).

By using post hoc tests, we measured the differences between the *T* and *SG*. When comparing the devices, it was obvious that the respondents selected the three lower evaluation levels (1 to 3) in the *T* vignettes less frequently than in the *SG* vignettes. Thus, by selecting the upper three levels (4 to 6), more than half of the participants were generally motivated to use *T* in everyday work, while less than half of them showed this motivation regarding *SG*.

In the next step, the Wilcoxon test proved significant differences. By combining the four vignettes from each device (*T* and *SG*) and comparing these two “new” variables, significantly more participants felt addressed by the *T* than by *SG* (Z = −6.653, *p* < 0.001). The effect size here was in the medium range (r = 0.385). *T*, as a technology with a low degree of innovation, was, therefore, associated with a higher motivation to use them than *SG*, supporting **H1**.

We also applied the Wilcoxon test for the other two dimensions. The results indicated no significant difference between all of the *EM* vignettes and all of the *IM* vignettes. We found a significant difference between the *PO* vignettes and the *EO* vignettes: *EO* increased the motivation to use digital innovations more than *PO* (Z = −2.995, *p* = 0.003), although the effect size was small (r = 0.173). We identified this significant difference in the characteristics of the value orientation, specifically in the *T* vignettes (Z = −2.182, *p* = 0.029) but also with a low effect size (r = 0.126). The medians of all of the individual expressions reflected these differences. They were mostly in the upper half of motivation to use (4). Those for the *PO* vignettes and the *SG* vignettes were in the lower half (3) (Table 4).

We only detected a significant difference within the motive dimension in the *SG* vignettes. Targeted pairwise analyses using the Wilcoxon test indicated significantly higher values for *EM* than for *IM* when we compared the combinations *SG-IM-PO* and *SG-EM-PO* (Z = −3.023, *p* = 0.002). However, the effect size was small (r = 0.16). For this reason, we could not confirm the assumption in **H2** that *IM* stimulated the respondents more to use digital innovations than *EM*, even if the descriptive values of the *T* vignettes initially suggested this. We also could not ultimately draw definite conclusions regarding the motives since we found differences only in combination with the other dimensions and their expressions.

We were able to disprove the assumption from **H3**. *EO* triggered higher motivation than *PO*, albeit a slight difference. Overall, the dimension of the degree of innovation had the highest influence on the motivation of nursing staff to use new technologies, followed by the value orientation, supporting **H4** and **H5**. Further analyses, with the combinations considered in isolation, did not yield any new findings for all of the dimensions and characteristics.

#### Influence of Independent Variables

We used group comparisons, regression analyses, and correlation calculations to measure the possible influences of the independent variables on the participants’ assessments of the eight vignettes. Five independent variables showed significant results across all of the vignettes. To measure the effect, we chose the vignette *T-IM-EO*, which had the highest ratings in our study. The higher the respondents’ general willingness to use technology (B = 0.289, *p* < 0.006) and the level of digitization of the hospital (B = 0.215, *p* < 0.011), the more likely the nurses felt addressed by the nursing leader’s announcements in all of the vignettes.

We found an equally positive correlation regarding the importance of efficiency in everyday work: The more important the efficiency was to the participants, the higher their motivation was to use *T* and *SG* in the fictitious situation (B = 0.171, *p* < 0.015). The present use of similar digital documentation, as with *T*, had a negative influence on the dependent variables in the eight vignettes (B = 0.502, *p* < 0.007). We found another negative correlation between the vignettes and the work experience at the current workplace (B = −0.037, *p* < 0.004). This means that the shorter the period of time for which the nursing staff had worked at their current workplace, the higher their motivation to use new technology.

## 4. Discussion

This vignette study identified motives and values that nursing leaders can focus on to motivate their teams when implementing digital innovations. Based on previous research and adjusting for context through consultation with nurses and nursing leaders, we integrated features with high motivational potential into fictitious situations that 299 hospital nurses then evaluated. The comparison of the two devices indicated that the motivation to use *T* was higher than the motivation to use *SG*. The challenge of implementing a higher level of innovation asks for even more focus on the motivational aspects. Furthermore, the participants felt more motivated when the nursing leader emphasized *EO* instead of *PO* in the announcement. With regard to the type of motivation, there did not seem to be significant differences between *IM* and *EM.* Thus, there was no “best” constellation presented. 

However, there was a high general willingness to use technology among a large proportion of respondents. The optional final comments in the survey reflected this: In contrast to findings in the literature, there was no general rejection of digitization in everyday work [56]. Most nurses were aware of the need to expand their competencies due to the process changes, whereby they saw digitization as a supportive opportunity in their work. They expected process optimization, especially in digital documentation, because they could access and exchange data quickly. This confirmed the importance of *EO* for the participants and their assessments, especially in the *T* vignettes. Furthermore, it supports the fact that new technologies need to be valid solutions for problems and challenges in nursing care. Just because digital opportunities exist and are new, they might not always be a better solution. Regarding this, responsible people in hospitals and health professionals, in general, should be critical, especially towards companies selling innovative digital products, in order to choose an option that they want to use in the long term.

The fact that many systems and products are not user-friendly or do not function properly was seen as critical. This was associated with additional work and disadvantages for the care itself and the patients. At best, digitization was associated with faster documentation and multi-professional exchange but barely with more time for patients. This may also explain why the respondents selected comparatively low values for the *PO* vignettes. For the nursing staff, there was probably no connection between *PO* and digitization anyway. That is why the respondents did not perceive this value orientation as motivating in the vignettes. However, *PO* provided significant value for nurses, as also confirmed in the survey.

Another explanation is that *EO* is crucial, especially in the context of digitization. Nursing staff that experience productivity and a good workflow are more satisfied and engaged, and vice versa [24,57,58]. Accordingly, *PO* and high-quality care for patients are only possible by working efficiently [1,59]. Without features such as efficiency and structure, the daily work routine becomes chaotic. Nursing leaders must focus on these factors in order to reduce the burden experienced by their staff and support their commitment and willingness to learn, especially in current pandemic times, and to face constant changes and innovations [39,60,61].

The negative influence of the existence of comparable digital patient documentation supports the assumption that the type of digitization that has existed so far does not work without disruptions and that there is a lack of training and support. More years of work in the current workplace also correlated with less motivation to use *T* and *SG*. The negative experiences could outweigh the positive ones, and generally poor working conditions lead to dissatisfaction and less engagement and motivation. The experienced disadvantages and the frequently reported low job satisfaction supported this [62,63,64].

A further explanation for the higher motivation in the *T* vignette is the association with efficiency: Disregarding the negative influences, *tablets* are well-known technologies that might not require as much training effort as *smart glasses*. A potential problem of the *SG* vignette could be the distance from reality and from being part of everyday practice. The lack of motivation of the nursing staff did not have to be primarily due to the technology itself, as the preparatory step of the consultation with nursing experts suggested. Its supportive, flawless use is still unrealistic, as the final comments of the participants confirmed. The nurses maybe could not see any work facilitation by using *SG*, which is why *EO* did not motivate them in the *SG* vignettes.

In addition, many people prioritize their private life and personal issues over their working life. The change of culture and values can therefore be another reason for striving for as much efficiency at work as possible so that the nurses can achieve a good work/life balance. The respondent group is young, well-educated, and attaches much importance to their own well-being and personal aims. Regarding *PO* in the *SG* vignettes, the respondents might not perceive any connection to *PO* but the idea of *SG* as a disruption of interpersonal communication. A possible explanation for the slightly higher motivation regarding *EM* in the *SG* vignettes could be perceived pressure to use. This may also have been a coincidence or a consequence of the complex vignette query.

Finally, nurses see high potential in digitization to facilitate work processes. They still cannot reconcile their hopes and, certainly, the existing acceptance with what has not yet succeeded in German hospitals [65]. The independent query on the importance of the motives and values at the end of the survey indicated the generally high importance of the expressions (*IM/EM/PO/EO*) in hospital nursing. The respondents classified *PO* as the most important, while this had comparatively low ratings in the vignettes. This suggests that the expressions were either not visible in the vignette formulation or that *PO* was simply not perceived as an advantage of using digital innovations in hospital nursing. Likewise, social desirability might have influenced the independent query, which the vignette query avoided.

Responsible nursing leaders should reflect the examined characteristics (*IM/EM/PO/EO*) in their behavior. In the context of digitization, however, there are far more problems that need to be fixed. These do not just relate to the staff but are primarily of a structural and organizational nature [66]. Leadership behavior remains influential [18,67]. Those who are responsible for the entire management and health policy levels must initiate the framework conditions. Regarding this, works in comparably developed countries, such as Canada, the USA, or England, can be of support [9,10,12]. The general conditions, such as the shortage of skilled nurses and pandemic-related additional burdens, also have an influence on the motivation of nursing staff, as reflected by the respondents’ moderate assessments of general job satisfaction. This induces a need for comprehensive action to ensure the willingness to learn and adapt, commitment, and, subsequently, high performance in nursing staff [60,63,64,68].

### Limitations

Despite significant differences, it is possible that respondents could not clearly distinguish and rate the vignettes. The frequent ratings of the medium levels (3 and 4) on the Likert scales support this assumption. Considering the dependent variable of motivation to use *T* and *SG*, a question with only “yes” and “no” as answer options could have generated pressure to make a decision and lead to rather negative evaluations. Even though we tested the content and wording for the dimensions and their expressions, understanding, and interpretation were not always the same, and respondents focused on different concepts. Otherwise, the combination of different characteristics in a general situation is the special character of the vignette design.

It is difficult to assess whether entirely different dimensions or different wording for the expressions would have been better. More situational features in the vignettes would have increased the complexity and, thus, reduced the respondents’ attention or even the overall response rate. Even though we collected many other factors influencing the motivation to use digital innovations in hospital care in the questionnaire, it is not possible to cover all of the influencing factors. Attitudes towards digitization are a causally multifaceted phenomenon, which is why our own survey could only explain a part of the motivation to use.

The higher proportion of female respondents in the sample gave the representativeness of the sample with regard to gender [14]. Limitations exist due to the random sample of the respondents. Recruitment via social and digital media led to self-selectivity in participation. The sample featured a young and educated group. The analysis, therefore, was controlled for the influences of such respondent characteristics, but we found no significant effects on the vignette ratings. The results are not representative of all German hospitals and their nursing staff, but this is not the prior aim of the qualitative research.

Finally, the external validity is limited, especially with regard to *SG*. Due to the participants’ possible lack of imagination, it was difficult for them to assess their behavioral intentions in the vignettes. Although the vignettes describe fictitious situations and inquire hypothetically about behavior, closeness to reality is still necessary. This seemed to be easier with the *T* vignettes. About half of the surveyed nurses already used this or a comparable technology in their everyday working lives; even more, they did so in their private lives. Whether the ratings ultimately match the behavior in a similar real-world situation remains open. The effect sizes in the statistical analysis were low to medium, but this is common in experimental designs.

## 5. Conclusions

Overall, the participants of our survey assessed the vignettes differently. The product design and degree of innovation had the biggest influence: the motivation to use was significantly higher for *T* than for *SG*. Even if the motives, *IM* and *EM*, as well as the values *EO* and *PO*, played a role in the work behavior of nurses, they were less relevant in the context of digitization. *EO* seemed to be at least more of an incentive to use digital innovations in everyday work than *PO*. Despite controlling for various influencing variables, the query was very complex, and we could not exclusively apply the results to general practice. However, they substantiate the relevance of communication skills in hospital nursing teams. Continuing education, workshops, and training on digital competencies and leadership communication should be designed to build up expert knowledge, pool experience, and foster personal growth for nursing leaders. Our results affirm that nursing staff generally display a high willingness to use technology. Products that seem unrealistic and unknown should find ways into everyday practice so that health professionals become familiar with them [1,25,61]. As numerous digitization projects have been and are still being launched, rigorous implementation research might offer further insights into the success factors. Taking into account the physical and psychological stresses on nursing staff that are increasing due to the pandemic, the results highlight the relevance of change management and a leadership-related urgency to maintain commitment and motivation among hospital nurses [60,68].

## Figures and Tables

**Figure 1 ijerph-19-10775-f001:**
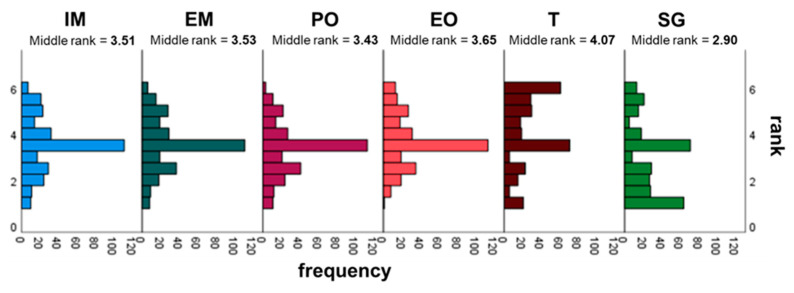
Two-factor variance analysis for ranks, according to Friedman, for the isolated expressions.

**Table 1 ijerph-19-10775-t001:** Baseline characteristics of participants.

Variable	N (in Total = 299)
**Gender**	
Male	69 (23%)
Female	229 (77%)
Diverse	1 (0%)
**Age (** $\bar{x}$ ** = 37 years)**	
Under 20 years	4 (1%)
20–29 years	95 (32%)
30–39 years	78 (26%)
40–49 years	59 (20%)
50–59 years	53 (18%)
Over 60 years	10 (3%)
**Highest level of education**	
Lower secondary school diploma	2 (1%)
Secondary school diploma	82 (27%)
General qualification for university entrance	112 (38%)
(Technical) College degree	100 (33%)
Not specified	3 (1%)
**Professional group**	
Health care and nursing	240 (80%)
Health care and pediatric nursing	41 (14%)
Nursing assistance	3 (1%)
Geriatric nursing	5 (2%)
Academic nursing degree	69 (23%)
Education	8 (3%)
Other	9 (3%)
**Specialization/further education**	
Yes	65 (22%)
No	234 (78%)
**General job satisfaction**	
Very low (1)	1 (0%)
Low (2)	12 (4%)
Rather low (3)	115 (39%)
Rather high (4)	126 (42%)
High (5)	30 (10%)
Very high (6)	15 (5%)
**General technical readiness**	
Very low (1)	0 (0%)
Low (2)	2 (1%)
Rather low (3)	24 (8%)
Rather high (4)	125 (42%)
High (5)	118 (39%)
Very high (6)	30 (10%)
**Frequency of T/SG use (private)**	
Never (1)	18/284 (6/95%)
Rarely (2)	30/1 (10/0.5%)
Sometimes (3)	17/4 (6/1%)
Often (4)	19/5 (6/2%)
Very often (5)	14/4 (5/1%)
Daily (6)	201/1 (67/0.5%)

**Table 2 ijerph-19-10775-t002:** Baseline characteristics of the respondents’ workplaces.

Variable	N (in Total = 299)
**Sponsorship**	
Public	213 (71%)
Non-profit/denominational	55 (18%)
Private	14 (5%)
Do not know exactly	17 (6%)
**Number of beds**	
Less than 100	6 (2%)
Less than 200	21 (7%)
Less than 300	6 (2%)
Less than 400	8 (2.5%)
Less than 500	22 (7.5%)
Less than 600	20 (6.5%)
Less than 700	11 (3.5%)
Less than 800	6 (2.5%)
800 or more	176 (59%)
Do not know exactly	23 (7.5%)
**Degree of digitization hospital/specialty department**	
Very low (1)	26/34 (9/11%)
Low (2)	46/55 (15.5/18%)
Rather low (3)	75/71 (25/24%)
Rather high (4)	82/65 (27.5/22%)
High (5)	57/57 (19/19%)
Very high (6)	13/17 (4/6%)
**Presence of digital documentation (T or similar)**	
Yes	162 (54%)
No	137 (46%)
**Frequency of T/SG use (professional)**	
Never (1)	147/291 (49/97%)
Rarely (2)	10/0 (3/0%)
Sometimes (3)	16/2 (5.5/1%)
Often (4)	21/3 (7/1%)
Very often (5)	29/2 (10/1%)
Daily (6)	76/1 (25.5/0%)
**Gender of leader**	
Male	182 (61%)
Female	117 (39%)
**Age of leader (** $\bar{x}$ ** = 47)**	
20–29 years	5 (1.5%)
30–39 years	56 (18.5%)
40–49 years	89 (30%)
50–59 years	110 (37%)
Over 60 years	33 (11%)
Not specified	6 (2%)

**Table 3 ijerph-19-10775-t003:** Importance of job characteristics for respondents.

Variable	Median	Mode
**Intrinsic motivation**	6	6
(“Interesting activities”)		
**Extrinsic motivation**	5	5
(“Good opportunities for advancement”)		
**Patient orientation**	6	6
(“Time for individual patients”)		
**Efficiency orientation**	6	5
(“Quick completion of tasks”)		

**Table 4 ijerph-19-10775-t004:** Vignette assessments: mode and median (individual and isolated per expression).

Variable	Median	Mode
T-IM-PO	4	4
T-IM-EO	4	4
T-EM-PO	3	3
T-EM-EO	4	3
SG-IM-PO	3	3
SG-IM-EO	3	3
SG-EM-PO	3	4
SG-EM-EO	3	3
T in total	4	4
SG in total	3	4
IM in total	4	3
EM in total	4	4
PO in total	3	3
EO in total	4	3

## Data Availability

The datasets used and/or analyzed in the context of this study are available from the corresponding author upon reasonable request.

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
