# Peer review of "Digitization in Everyday Nursing Care: A Vignette Study in German Hospitals"

_ijerph, 2022, doi:10.3390/ijerph191710775_

Round 1
Reviewer 1 Report (Previous Reviewer 2)
Dear authors, thank you for resubmitting your manuscript. I have no further comments except encouraging you to perform an English spell check.
Author Response
Thank you for your positive feedback as well as your advice. We read the manuscript several times in detail and changed some sentences, e.g. from passive to active. An English professional also checked the revised manuscript, which an English editing service checked before.
Reviewer 2 Report (New Reviewer)
Thank you so much for the effort to update the manuscript. However, I believe there is some scope to update the related or previous state of work in the introduction section, or authors can add a separate section for related work. And they should add the most recent work done in this area. For example:
"Arefin, Kazi Zawad, et al. "Towards Developing An EMR in Mental Health Care for Children’s Mental Health Development among the Underserved Communities in USA." 2021 IEEE 45th Annual Computers, Software, and Applications Conference (COMPSAC). IEEE, 2021."
And so on.
Author Response
Thank you for your feedback and the valuable notes. According to your advice, we added new literature and mentioned related current work like the example you gave. The sections were integrated in the introduction and are now as follows:
“Digitized documentation processes reduce paper work, create optimized schedules and increase productivity, allowing nurses more time for patient care (7-10). Electronic medical records encourage the health professionals´ exchange and decision-making (11, 12). Another digital opportunity to support patient safety and quality of care is video consultation (13) which nurses in hospitals can perform e.g. with tablets.” (line 37-42)
“Likewise, both devices can support nurses and increase patient safety when nurses use them in context of video consultation (9, 13). Smart glasses are also promising in con-text of nursing education. For instance, they can increase learning attitude and motivation of nursing students as well as support the students´ learning and comprehension (28, 29).” (line 65-69)
This manuscript is a resubmission of an earlier submission. The following is a list of the peer review reports and author responses from that submission.
Round 1
Reviewer 1 Report
Dear authors,
Thank you for the performed work. I am very sorry, but the aims and methods of the study and thus the evaluation data do not come clear to me.
“In this respect, leaders can motivate nurses.” Who is meant by “leaders” – this is a major issue throughout the complete document. It does not become clear to me who is addressed by the study. Please clarify.
“What motives and values can they address to support nurses’ motivation to use different digital innovations? We selected a tablet (T) as a classic device with a low degree of innovation and smart glasses (SG) as a future-oriented digital solution mainly existing in the research context. “ What are these devices used for?
Methods
“The situations differed regarding the devices (tablet/smart glasses), addressed motives 18 (extrinsic/intrinsic), and values (efficiency/patient orientation).”
It does not become clear to me what the given situations and vignettes really included and addressed. That should be situations that are relevant to the “leaders”? Or the nurses in their daily life? As the content of the vignettes is not described very well the complete evaluation is of low value to me. What is the content difference between T and SG vignettes (besides the hardware)?
How did the authors measure extrinsic/intrinsic motivation – does not become clear to me.
The explanations on why vignettes are of methodological advantage can be shortened.
Result section
Major parts of the text e.g. line 122-132 is then completely given in the tables again. This is redundant information. As the manuscript is rather long it can be condensed.
Table 1 Numbers of males and females seem to be mixed up.
The authors should decide whether they report der percentages with or without decimals. I would recommend without.
General job satisfaction and technical readiness are reported together in table 1 – these are two different issues and should be reported separately.
Mean values e.g. lines 154 and 164 are not the appropriate numbers to report in an ordinal variable
Discussion
“In summary, this vignette study identified ways in which hospital nursing leaders 233 can motivate their staff to use digital innovations.” Please clarify the “identified ways”.
Reviewer 2 Report
Dear authors, I read the article entitled with great interest. Readers might profit from your article. However, I have some minor suggestions for the revision of the manuscript and I encourage you to re-submit your article.
Introduction:
Please consider to provide more details on the problem in healthcare which is addressed. For example, why is the potential of digitization high? In addition, please elaborate on the potentials of digitization in healthcare.
Further please provide some more details on the systematic literature research and interviews with experts you have conducted before.
Overall, please consider rephrasing some of the sections in the manuscript. Keeping it simple might help here. In addition, please consider to consult a native speaker about academic writing.
Conclusions:
Please consider to include more recommendations for stakeholders in health care.
Reviewer 3 Report
I would like to congratulate the authors of this study, and I would like to express my gratitude. Sometimes are forgotten topics to research and it is necessary to point them.
Only I have considered to authors and editor: in the methods section, maybe the presentation of the hypothesis could be highly academic and rigorous in the written mode. If the journal is accepted this way, happy with this.
Best regards